# Polytypes of sp²-Bonded Boron Nitride

Bernard Gil [1,*], Wilfried Desrat [1], Adrien Rousseau [1], Christine Elias [1], Pierre Valvin [1], Matthieu Moret [1], Jiahan Li [2], Eli Janzen [2], James Howard Edgar [2] and Guillaume Cassabois [1]

1 Laboratoire Charles Coulomb, UMR 5221 CNRS-Université de Montpellier, F-34095 Montpellier, France; wilfried.desrat@umontpellier.fr (W.D.); adrien.rousseau@umontpellier.fr (A.R.); christine.elias@ens-paris-saclay.fr (C.E.); pierre.valvin@umontpellier.fr (P.V.); matthieu.moret@umontpellier.fr (M.M.); guillaume.cassabois@umontpellier.fr (G.C.)
2 Tim Taylor Department of Chemical Engineering, Kansas State University, Manhattan, KS 66506, USA; jiahanli@ksu.edu (J.L.); elijanzen@ksu.edu (E.J.); edgarjh@ksu.edu (J.H.E.)
* Correspondence: bernard.gil@umontpellier.fr

**Abstract:** The sp²-bonded layered compound boron nitride (BN) exists in more than a handful of different polytypes (i.e., different layer stacking sequences) with similar formation energies, which makes obtaining a pure monotype of single crystals extremely tricky. The co-existence of polytypes in a similar crystal leads to the formation of many interfaces and structural defects having a deleterious influence on the internal quantum efficiency of the light emission and on charge carrier mobility. However, despite this, lasing operation was reported at 215 nm, which has shifted interest in sp²-bonded BN from basic science laboratories to optoelectronic and electrical device applications. Here, we describe some of the known physical properties of a variety of BN polytypes and their performances for deep ultraviolet emission in the specific case of second harmonic generation of light.

**Keywords:** boron nitride; polytypism; deep ultraviolet emission

## 1. Introduction

Developing electronics capable of operating at high frequency and high power, at high temperature, and in harsh environments and optoelectronics with absorption/emission with wavelengths shorter than 400 nm requires the global control of the different branches of the technology. To date, research has focused on some group IV semiconductors such as diamond or silicon carbide and the group III element nitrides. The now fairly mature technology of nitride semiconductors extensively uses heterostructures of (Al,Ga,In)N solid solutions with atoms stacked according to the $C_{6V}$ hexagonal (wurtzite) symmetry [1]. After some attempts, the prospect for producing commercial devices based on the cubic (zincblende) polymorphs of these alloys has been almost abandoned, because it is difficult to synthesize these materials with qualities as good as those with the wurtzite structure [2,3]. For deep UV range of emission/absorption, i.e., light with wavelengths near 200 nm, aluminum nitride and boron nitride (BN) are, among the III–V compounds, both candidates of choice [4]. Concerning the family of II–VI compounds, magnesium-rich solid solutions of (Zn,Mg)O will be probably used. Each of these three candidate semiconductor systems share similar severe difficulties, such as the control of their electronic properties by impurity doping. There are also more specific ones: a wurtzite to rock-salt phase transition versus Mg composition for the II–VI compounds alluded to earlier [4,5], or polytypism for boron nitride. All these variations in the crystal structure impact charge carrier mobilities and diffusion lengths and the interaction of electromagnetic fields with the electronic states (optical internal quantum efficiency). In this article, we focus on the polytypism in sp²-bonded boron nitride, a layered two-dimensional compound similar to graphite, mica, $MoS_2$, or InSe. The different polytypes are associated with different space groups and different physical properties such as the occurrence or not of piezoelectricity, ferroelectricity, spontaneous polarization and second harmonic generation. The choice of this material is linked

to its technologically demonstrated interest and to its optical properties: stimulated light emission at 215 nm was demonstrated in single crystals by cathodoluminescence as early as 2004 [6], and an electrically-driven field emission device was later demonstrated [7]. Boron nitride is also a developing material for quantum technologies, specifically single photon sources operating in the UV from topological defects of the crystal. Such quantum emitters have not been reported in classical nitrides, although it has been already demonstrated in hBN with operation at 4 eV [8]. It was recently reported by a bottom–up growth technique at the UCB (University of California, Berkeley, CA, USA) that slightly twisted two-layered BN crystals with reduced symmetry could dramatically enhance the intensity of this emission [9]. A various series of applications have also been proposed regarding nanophotonics with hBN, which are partly reviewed in [10].

There exists no theory about the origin of polytypism in stackings of arrays of six-ring made alternating boron and nitrogen atoms. A case by case analysis is required for the interpretation of the origin of this phenomenon in a given compound [11]. In SiC, ZnS, $CdI_2$, or micas, this has been already understood, based on a sophisticated application of advanced thermodynamic models and dislocations-assisted growth mechanisms [12–14]. These studies demonstrate that there does not exist a universal model to describing the origins of polytypism and to quantitatively anticipate its consequences on the physical properties of BN crystals out of campaigns of experimental and theoretical investigations.

## 2. The Many Shapes of BN: Polymorphism and Polytypism

Boron nitride was synthesized as early as 1842 [15], and it appeared as a white powder made of small flakes revealed by optical microscopy. Advantage was taken of its high melting temperature (above 2900 °C) for hot-pressing this powder in order to design specific forms (crucibles for the growth of crystals for instance) for industrial operations in high-temperature conditions. Its powders could be used as a dry solid lubricant, analogous to black graphite or $MoS_2$. The efficient interaction of neutrons with the nucleus of the $^{10}$B isotope, which is present with $^{11}$B with respective proportions of 20/80 in natural boron, was also evidenced [16–18]. Adding it to concrete can provide some protection to neutron irradiation. These physical properties paved the way for interesting industrial applications, including cosmetics, a couple of decades before the birth of solid-state electronics. The crystalline structure of such BN powders leads to controversial suggestions [19–22] until R.S. Pease [23] clearly proposed (for the hBN powders, he was studying the X-ray diffraction features), an $sp^2$-type of chemical bonding. The easy gliding of the (001) crystallographic planes under shear stress is prototypical of layer compounds including hBN, graphite and $MoS_2$. These and many other layered compounds typically have strong chemical bonds in the plane of the layers and weak van der Waals bonds orthogonal to the planes. Thus, van der Waals interactions rule the stability of the stacking in the ⟨001⟩ direction. Recently, Vuong et al. [24,25] extracted the electronic distribution around nitrogen and boron atoms by studying the X-ray diffraction features over the full reciprocal space of high-quality hBN single crystals. The electron density is significantly higher around nitrogen atoms than boron atoms as theoretically predicted [26], with in addition, some slight differences around $^{10}$B or $^{11}$B isotopes in case of isotopically purified hBN [25]. The six-ring stacking that is derived from these studies is a perfectly (at the scale of the sensitivity of the x-ray experiments) ordered stacking of a honeycomb two-dimensional lattice of six-ring layers, with perfectly aligned lines of alternating boron and nitrogen atoms from one plane to another, along the ⟨001⟩ direction [23]. Such three-dimensional arrangement of the boron and nitrogen atoms follows the $D_{6h}^4$ (also noted $P6_3/mmc$ or N°194) space group symmetry, and it is sketched in along the ⟨001⟩ direction of the hexagonal lattice in Figure 1a. Hexagonal boron nitride is the most common BN polytype, but it is not the only one: far from it [27–32].

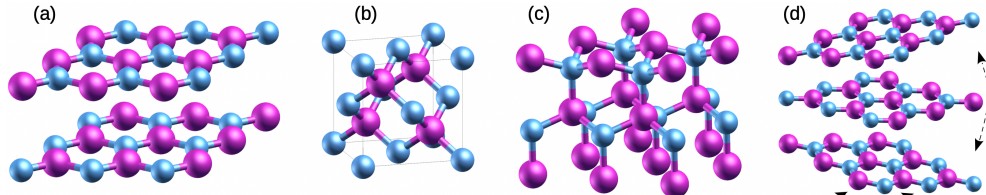

**Figure 1.** BN polymorphs. The stacking of boron (magenta spheres) and nitrogen (blue spheres) atoms for (**a**) the sp$^2$ bonded hBN, and sp$^3$ bonded (**b**) zinc blend, and (**c**) wurtzite polymorphs and (**d**) the turbostratic disordered stacking. The vertical axis is the ⟨001⟩ crystallographic direction for all the structures.

## 2.1. Polymorphisms

Allotropy (in case of a single element) and polymorphism (for a pure compound), which are frequent phenomena, are two very much documented experimental and theoretical branches of materials science. In 1788, Martin Heinrich Klaproth has revealed that rhombohedral calcite and orthorhombic aragonite were astonishingly two polymorphs of calcium carbonate CaCO$_3$. Two centuries before fullerenes and carbon nanotubes were identified, Smithson Tennant and William Hyde Wollaston demonstrated in 1796 that diamond was an allotrope of carbon with cubic symmetry. These demonstrations were very exciting for contemporary scientists, and they are the origins of the science of polymorphism and polytypism in BN crystals discussed here. It took about one century after Balmain grew sp$^2$-BN for achieving the growth of boron nitride crystals with four-fold coordinated atoms (sp$^3$ hybridizations of the atomic states) with cubic (zinc blende) symmetry. It required a pressure-induced (at about 4.5 GPa) polymorphic phase transition at 1500 °C of sp$^2$-bonded boron nitride powders [29,33,34]. The atomic stacking of boron and nitrogen atoms follows the T$_d^2$ (also noted *F$\bar{4}$3m* or N° 216) space group symmetry [35], which is sketched in along the ⟨001⟩ direction of the cubic lattice in Figure 1b. Studies were launched to establish the complete phase diagram of boron nitride [36–39], in the exciting high-temperature (and eventually pressures) ranges required to induce the phase transitions between the different polymorphs. The cubic polymorph cBN exhibits a super high hardness (second only to diamond), thus motivating its main use in abrasive applications [40]. With its large bandgap and high thermal conductivity, cBN is a potential challenger of other wide bandgap semiconductors such as SiC and GaN, for electronic operations under extreme conditions [41]. Under high-pressure and high-temperature conditions, a further polymorphic phase transition toward wurtzite BN can be also achieved [42–46]. The number of atoms per shells of successive neighboring (first, second, …) can be very different for the polymorphic structures of a given compound: for instance, four in the case of the zincblende polytype, to six for the rock-salt, and eight for CsCl. The atomic stacking of boron and nitrogen atoms which follows the C$_{6v}^4$ (also noted *P6$_3$mc* or N° 186) space group symmetry is sketched in along the ⟨001⟩ direction of the hexagonal lattice in Figure 1c. More recently, BN nanotubes were also grown [47,48].

## 2.2. Polytypisms

In addition to the structure variation with sp$^3$-bonded cBN and wBN, sp$^2$-bonded BN polytypes are also possible. Polytypism is a crystallographic property found with many mineral and organic crystals with compact or layered structures. Crystallographers were the first to investigate the occurrence of a large number of structural varieties in specific compounds. The best example of a material with many polytypes is silicon carbide. Its polytypism was discovered early [49,50], and this triggered pioneering theoretical studies to elucidate its origin, some based on thermodynamics [51] and others invoking specific growth mechanisms such as the defect-assisted spiral growth [52]. Chemists, physicists and engineers became interested in this, as it offers many fundamental issues to disentangle. Some physical properties such as: hardness and density remain almost unchanged between SiC polytypes. Other properties are significantly different. Piezo-electricity vanishes

with the occurrence of inversion symmetry, pyroelectricity can exist or not, charge carrier mobilities and optical properties are all sensitive to the polytype. The ability to control the polytype at will would enable the tuning of these properties. If the elementary cells of these atomic stackings share two lattice parameters, whilst the third value changes from one polytype to another one, the polytypism is called monodimensional polymorphism. There exists some cases where the value of this third parameter is not changed; in that case, polytypes are distinguished by different structures of the crystalline pattern. The structures of the different polytypes of a given compound can be described using different Bravais lattices and can belong to different space groups. The structural patterns of the polytypes, whether they are compact three-dimensional compounds (such as SiC, ZnS) or layered compounds (such as mica, BN, graphite, Transition Metal Dichalcogenides, etc.) can always be described in terms of different stackings under the vertical direction of an invariant generic stacking. In the case of some compounds, there also exists a fully disordered stacking structure for which the periodicity vanishes along the normal to the layers [53]. There are many possible intermediates between aperiodic and periodic stackings. The most interesting are these based on periodic stackings flanked by a variable amount of disordered stacking faults. This is known as the turbostratic stacking, tBN, which is sketched in Figure 1d [54]. Among the polytypes, the macroscopic shapes of the crystals do not change in contrast to the case of polymorphs.

Figure 2 contains plots of the different ordered atomic stackings described in this review. In the AA stacking of Figure 2a, the six rings of alternating B and N atoms are perfectly stacked along the $c$ direction of the crystal, a given atom of a plane facing similar ones in the planes above and below. In the AA' stacking (also called hBN) in Figure 2b, the alternating planes experience a 60° twist from one to another, giving along the $c$ axis, atomic lines made of alternating B and N atoms. Assuming a model of rigid atomic planes, we can conclude that AA stacking is unstable, due to the nitrogen–nitrogen and boron–boron repulsive electrostatic interactions. AA' stacking is much more stable due to the attraction between boron and nitrogens in adjacent layers [11]. The staggered AB stacking displayed in Figure 2c, also called Bernal stacking, bBN, is obtained from the AA stacking by gliding the atomic plane of the upper layer to the center of the six rings of the lower layer in such a way that half of the boron and nitrogen atoms of adjacent planes face each other; the remaining others sit at the center of the interstitial voids of the honeycomb lattice. This structure is similar to the structure of graphite, and it is energetically more stable than AA [11]. There are two other staggered stackings. In one of them, called $AB_1$ here, the gliding of one plane to the other one is similar to that in the AB case, but it occurs in such a way that the boron atoms of successive planes are facing each other whilst nitrogen atoms sit at the center of the interstitial vacancy of the hexagonal lattice (Figure 2d). Finally, we indicate the $AB_2$ stacking where the nitrogen atoms of successive planes are facing each other whilst boron atoms sit at the center of the interstitial vacancy of the hexagonal lattice (Figure 2e). After this description of possible bilayer stacking sequences, we discuss the possibilities of longer periodic stacking with more than two layers. This is the case of the rhombohedral stacking plotted in Figure 2f. Here, the gliding vector is smaller than it is for the AB structure, and two slidings are required to recover the crystalline periodicity, which we call ABC or rBN. We restrict to these polytypes our journey into the many possibilities, since the highest probability of occurrence of polytypes generally occurs for the smaller periodicities.

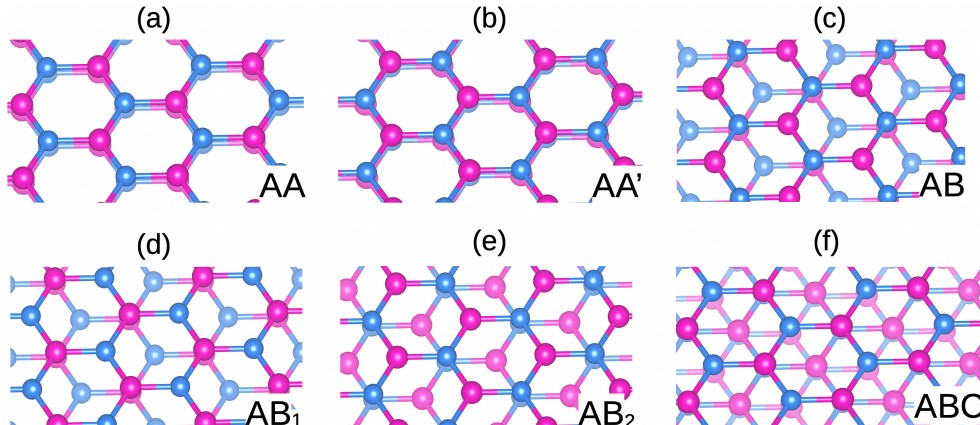

**Figure 2.** Top views of the sp² polytypes of BN with boron and nitrogen atoms indicated by magenta and blue spheres, respectively. The atomic stacking of the individual layers are: (**a**) AA, (**b**) AA' or hBN, (**c**) AB or bBN, (**d**) $AB_1$, (**e**) $AB_2$, and (**f**) ABC or rBN. The six stackings are represented as an artist view in the (001) plane slightly tilted.

### 2.3. Crystal Cohesive Energies and Values of the Lattice Parameters

In the previous section, we discussed the stability of the different BN stackings, founded on the rigid atom basic approach. With the aim at quantifying this stability, we have calculated the energy as a function of the volume data for nine cristallographic structures of boron nitride. The total energies of the different polymorphs have been computed with the quantum ESPRESSO code within the density functional theory, based on Perdew–Zunger local density approximation exchange–correlation potentials [55,56]. The crystallographic cells have been first relaxed before calculating the total energy as a function of the cell volume by varying the cell parameters. A *k*-point sampling of $10 \times 10 \times 10$ and a plane-wave cutoff of 100 Ry were used. The calculated total energies per atom are displayed as a function of the volume per atom in Figure 3. Our results are in overall agreement with the calculations of Refs. [26,57–65]. In addition, we present the result predicted for the rock-salt stacking, which matches very well to those published by Furthmüller et al. [58]. Last, many more different layered stackings are computed in ref. [61].

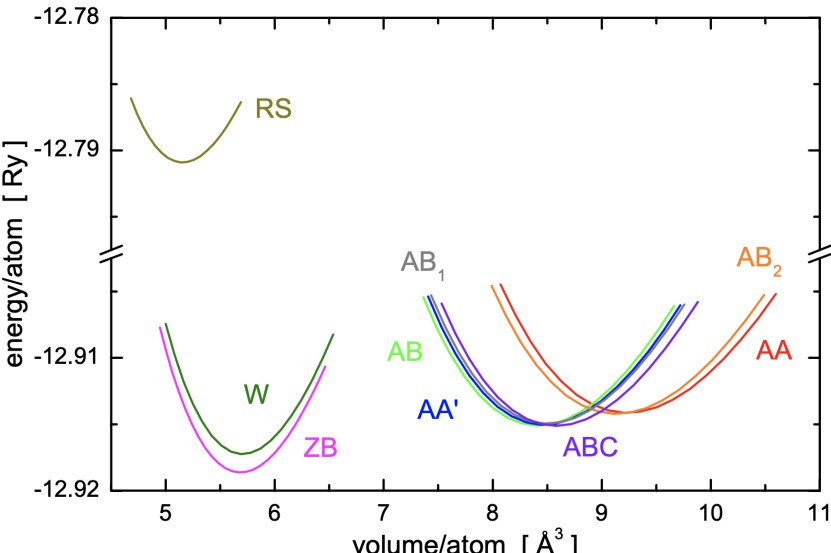

**Figure 3.** The energy per atom vs. volume per atom computed for nine different structures of boron nitride: rocksalt (RS), wurtzite (W), zincblende (ZB), and the six indicated polytypes.

The values of the relaxed lattice parameters are given in Table 1. In the last column, we reproduce the experimental parameters of ref. [66] that are of interest for comparing with the computed ones. Generally, the energies for AA′, AB and $AB_1$ stackings are very close. The energies for AA and $AB_2$ stackings are higher. While the in-plane boron–nitrogen bond lengths do not vary drastically, that is not the case concerning the values of the *c*-axis parameter. The interlayer spacings are globally similar for the AA′, AB, $AB_1$ and ABC stackings (evidently, the *c* parameter is 1.5 times larger for the rhombohedral stacking than for the two-layer periodic stackings), while the values of the AA and $AB_2$ stackings are significantly larger. Although the rigid atom model is far from exact and does not distinguish the different "sizes" of the B and N atoms as derived from quantum mechanics, it contains the spirit of the physics. The major difference between the $AB_1$ and $AB_2$ stackings relies on the larger electronic clouds of the nitrogen atoms in relation to the boron atoms. The N-N superposition in the AA and $AB_2$ stackings lead to strong repulsive interactions between adjacent layers and make these structures energetically unstable in relaxed conditions. The experimental values of the lattice parameters of the AA′ and ABC stackings are the only ones to have been accurately determined experimentally [23–25,67–70], and their values agree with the theoretical predictions.

**Table 1.** Theoretical and experimental values of the lattice parameters for nine Bravais lattices of boron nitride.

| Structure | Atoms facing along *c* (When Any) | Theory | | Experiment [66] | |
|---|---|---|---|---|---|
| | | *a* (nm) | *c* (nm) | *a* (nm) | *c* (nm) |
| Rocksalt | - | 0.3451 | - | - | - |
| Zincblende | - | 0.3566 | - | 0.3615 | - |
| Wurtzite | - | 0.2513 | 0.4159 | 0.2551 | 0.4210 |
| AA′ | N-B | 0.2478 | 0.6354 | 0.2504 | 0.6656 |
| AB | N-B | 0.2477 | 0.6319 | - | - |
| $AB_1$ | B-B | 0.2476 | 0.6384 | - | - |
| ABC | - | 0.2476 | 0.9679 | 0.2504 | 0.999 |
| AA | N-N and B-B | 0.2476 | 0.3468 | - | - |
| $AB_2$ | N-N | 0.2476 | 0.686 | - | - |

*2.4. X-ray Diffraction Spectra for the Different Polytypes*

In Figure 4, we plot the theoretical intensities of the X-ray diffraction (XRD) peaks of powders of the different ordered $sp^2$ stackings under discussion. In each case, given a polytype, the intensities of the XRD peaks are plotted relatively to the normalized intensity of the (00*l*) planes, with $l = 1$ for AA, $l = 2$ for AA′, AB, $AB_1$ and $AB_2$, and $l = 3$ for ABC. The diffraction angles are calculated from the values of the lattice parameters given in Table 1 for the relaxed cells, i.e., which minimize the total energy. The interesting point is that the values of the diffraction angles for the AA and $AB_2$ stackings are lower by approximately 2° than their analogs in the series of the $sp^2$ BN polytypes plotted in Figure 4. The vertical dashed line highlights the diffraction angle of the (002) peak in the case of the AA′ stacking. We note that the angle value of the turbostratic boron nitride, in which the stacking of the layers is irregular, is shifted by about 0.5°–1° with respect to the (002) peak of hBN [67,71,72]. In some cases, this value may be similar to the shift observed for the AA and $AB_2$ stackings, which could eventually lead to a misinterpretation of the structural nature of the epilayers.

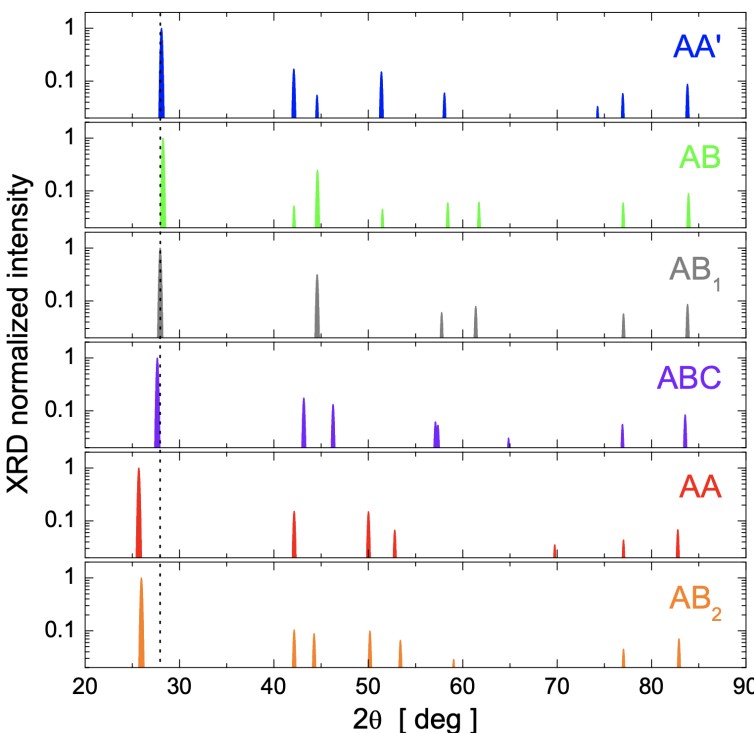

**Figure 4.** X-ray diffractograms computed for powders of sp²-bonded ordered BN stackings, based on the theoretical lattice parameters. The X-ray wavelength was taken equal to 1.541842 Å(Cu-K$\alpha$ line).

*2.5. Symmetries and Dispersion Relations of Phonons for the Different Polytypes*

The dispersion relations of phonons computed for the relaxed structures are plotted in Figure 5 along the $\Gamma - K$ path for the AA, AA' and ABC polytypes. At first sight, the three phonon band structures look similar, but significant differences exist. We have summarized in Table 2 some specificities of the phonon modes of the monolayer AA stacking ($D_{3h}$ point group), of the AA' and AB bilayer stackings ($D_{6h}$ and $D_{3h}$, respectively), and of the three-layer ABC stacking ($C_{3v}$). There are six, twelve and eighteen vibrational modes for the one, two and three-layer stackings. Thanks to the characters of the representation tables of the different groups, the modes present A-like symmetries (label A indicates vibrations of atoms that occur in the plane orthogonal to the layers), E-type symmetries (label E indicates a double degeneracy linked to vibrations in the plane of the layers), and $B_{1g}$ symmetries that correspond to silent modes in the case of the AA' stacking. Silent modes cannot be detected using an optical experiment, but they exist, and they can impact the diffusion of carriers. Of course, in all cases, there are three acoustic phonon modes. It is worthwhile noticing the lack of a low-energy mode of E-type symmetry, which is either Raman active or both Raman and IR active, in the 55–60 cm$^{-1}$ range and the existence of a Raman activity in the 800 cm$^{-1}$ range for the ABC stacking.

Therefore, each stacking reacts specifically to incoherent light scattering, and infrared absorption experiments can (at least theoretically) permit a structural diagnosis. As usual, the linewidth of the optical features will be decisive for achieving it. Since the pioneering work of Geick et al. [73], there have been many studies dedicated to phonon properties in hBN (AA') [74–84] due to its efficient potential for thermal management [77,85–92] or nanophotonics [93–98].

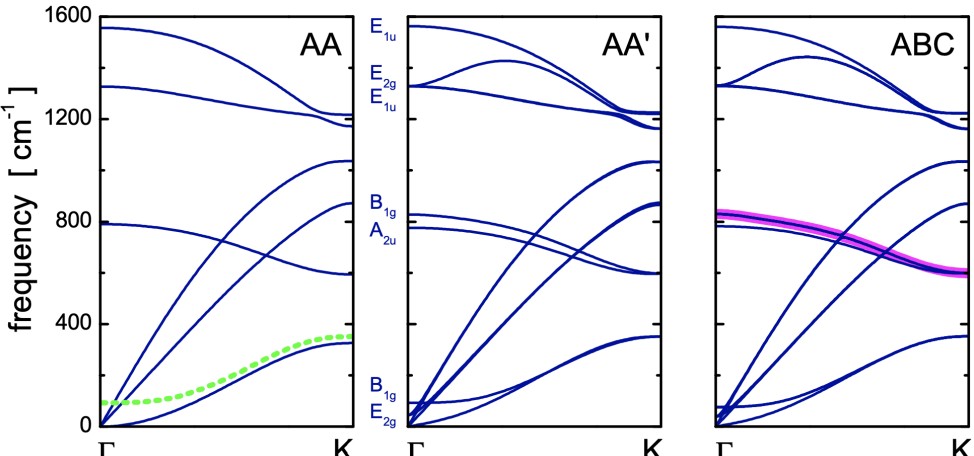

**Figure 5.** Dispersion of the phonons along $\Gamma - K$ for the AA (**a**), AA′ (**b**) and ABC (**c**) stackings of boron nitride. The green dotted line in (**a**) shows a mode which is absent in the AA stacking. The magenta curve in (**c**) stands for the additional Raman mode in the 750–800 cm$^{-1}$ range.

**Table 2.** Phonon modes specifications of four BN stackings.

| Stacking | Energy Range 1370 cm$^{-1}$ | Energy Range 750–800 cm$^{-1}$ | Energy Range 0–120 cm$^{-1}$ | Specific Items to Outline |
|---|---|---|---|---|
| AA | E′ (IR + R) | A″$_2$ (IR) | A″$_2$(ac) + E′(ac) | No vibration mode for Raman activity exists in the 0–120 cm$^{-1}$ range. Only acoustic modes. |
| AA′ | E$_{2g}$ (R) + E$_{1u}$ (IR) | B$_{1g}$ (S) + A$_{2u}$ (IR) | A$_{2u}$(ac) + E$_{1u}$(ac) + B$_{1g}$ (S) + E$_{2g}$ (R) | There are B$_{1g}$ silent modes. Raman and IR active modes have different symmetries. |
| AB | 2E′(IR + R) | 2A″$_2$ (IR) | A″$_2$(ac) + E′(ac) + E′ (IR+R) + A″$_2$ (IR) | The AA′ silent modes B$_{1g}$ become IR active (A″$_2$). E$_{1u}$ and E$_{2g}$ modes become simultaneously IR and Raman vanishing of inversion symmetry. |
| ABC | 3E (IR + R) | 3A$_1$ (IR + R) | A$_1$(ac) + E(ac) + 2(E + A$_1$)(IR + R) | Raman active modes in the 750–800 cm$^{-1}$ range. |

### 2.6. Polytypism in Optical Experiments of the Early Days

Since the beginning of their experimental investigations, the fluorescence spectra of the sp$^2$-bonded crystals were measured as broad bands of about 500 meV width at half maximum, peaking in the 5.75 eV range and accompanied with many additional contributions on the low and high-energy sides of the main peak. In fact, such measurements were performed very early on powders or nanocrystals. This complicated the interpretation of light emission by layered boron nitride for decades. Representative BN spectra can be found in Refs. [63,99–104]. Fluorescence spectra, produced by electrons excited by a highly energetic photon or electron beam (cathodoluminescence), give a snapshot of the radiative recombination mechanisms of carriers with populations ruled by thermalization processes between all their possible levels in the bandgap [105]. These spectra give an image of the crystal properties in terms of its extrinsicity and promote the low-energy radiative recombination energies.

Absorption or reflectivity experiments, in contrast, probe the intrinsic and direct (in reciprocal space) transitions [106]. The oldest of them can probably be attributed to Professor Wolfgang Choyke. He never published his experimental data himself, but he offered his result to Doñi and Parravicini [107] to help them improve the tight-binding description of the band structure of BN. Since that time, there have been several studies

dedicated to reproducing and improving his pioneering experiment. In Figure 6, we plotted some measurements of the dielectric constant of sp²-bonded BN recorded using different techniques (ellipsometry, absorption, photoconductivity) at different periods, with time passing. Spectra (a), (b) and (c) are experiments taken on pyrolytic BN powders [107–109]. Note the two main resonances: one at about 6 eV and another about 1 eV higher.

Other features, which cover narrower energy scales, were collected on bulk crystals [110] (d) and epilayers [111,112] (e) and (f). The trend in going from pyrolytic BN to bulk crystals via epilayers is found when performing a photoconductivity experiment on a bulk sample, as indicated by the spectrum (g) [113]. These results were published by different groups, and the scatter in the data cannot be attributed to measurement artefacts or at the different temperatures at which experiments were performed. However, in light of what is known today, this can be attributed to the different crystallographic structures of the samples investigated by the different groups.

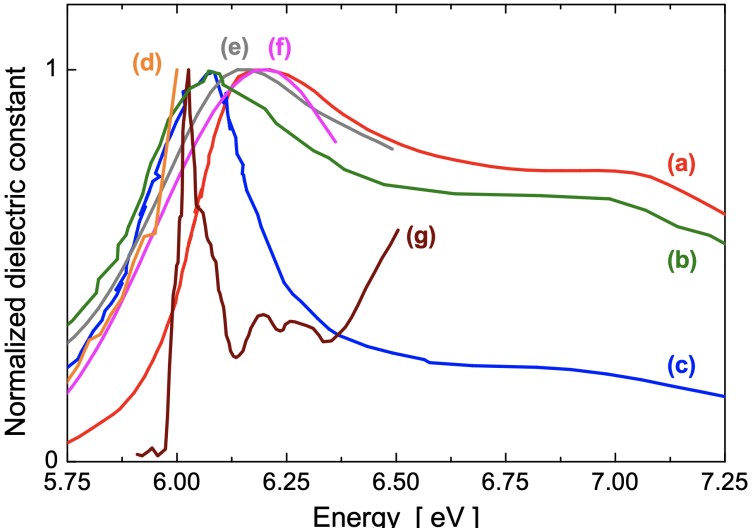

**Figure 6.** Dielectric constant vs. energy measured at room temperature on powders, bulk crystals and epilayers. Note the distribution of tendencies for the pyrolitic BN (**a–c**), the epilayers (**e,f**) and the bulk hBN crystals (**d,g**).

Arguably, the differences are linked to inhomogeneities of the strain states of the samples that were investigated. Under the application of a hydrostatic pressure, the direct bandgap of the AA′ stacking decreases of $26 \pm 2$ meV/GPa [114,115]. A compression of 1 GPa produces a variation of 2.41% of the $c$ lattice parameter [116]. This leads to a variation of the lowest of the diffraction angles of the $(00l)$ planes of about 0.8° toward high angles, which was not reported experimentally. Therefore, the changes of the resonance energies cannot be attributed to a hydrostatic stress effect, at least to first order. Regarding anisotropic stress that could be operating in the plane of the layer, we consider now a uniaxial stress $\sigma$ along an in-plane direction $x$ (see Figure 7), which we represent as follows in the Voigt notation [117]:

$$\sigma' = \sigma \begin{pmatrix} 1 \\ 0 \\ 0 \\ 0 \\ 0 \\ 0 \end{pmatrix} \tag{1}$$

It generates a strain field that writes:

$$\epsilon' = \sigma \begin{pmatrix} S_{11} \\ S_{12} \\ S_{13} \\ 0 \\ 0 \\ 0 \end{pmatrix} \tag{2}$$

The signs of the components of the compliance tensor $S_{11}$ and $S_{12}$ or $S_{13}$ are different, and the changes of the lengths of the vectors $x, y, z$ are [117]:

$$\begin{pmatrix} x(\sigma) \\ y(\sigma) \\ z(\sigma) \end{pmatrix} = \begin{pmatrix} 1+\epsilon_{xx} & \epsilon_{xy} & \epsilon_{xz} \\ \epsilon_{xy} & 1+\epsilon_{yy} & \epsilon_{yz} \\ \epsilon_{xz} & \epsilon_{yz} & 1+\epsilon_{zz} \end{pmatrix} \begin{pmatrix} x(0) \\ y(0) \\ z(0) \end{pmatrix} \tag{3}$$

In that case, the orientations of the crystallographic directions $x, y$, and $z$ are constant, but the unit cell lengths change. The symmetry of the crystal becomes orthorhombic and the table of character of its representative point group contains only real numbers. If a negative value is attributed to a stress compression along $x$, then $S_{11} > 0$ and $S_{12} < 0$. Using the notations of Figure 7, the length of $x$ decreases; therefore, the length $\Gamma K_1$ increases and the length of $\Gamma K_2$ decreases. The direct transition at the series of $K_i$ points of the Brillouin zone, plotted in green in Figure 7, splits into two (one for the valence and conduction states both at $K_1$ and $K_4$, another involving those of these states sitting at the remaining states sitting at $K_2$, $K_3$, $K_5$ and $K_6$). Theoretically, it results in a double structure or in a broadening or, depending on the value of the stress and of the deformation parameters of the conduction and valence band extrema at a given family of $K$s, in the natural broadening of the reflectivity feature. This simple description of the effect of a uniaxial strain inside a BN layer does not explain reasonably the qualitative differences observed in the experimental curves of the dielectric constant plotted in Figure 6.

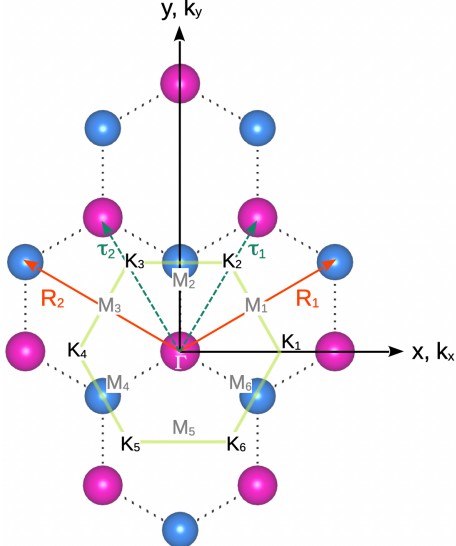

**Figure 7.** Atomic arrangement of boron (magenta) and nitrogen (blue) atoms in a monolayer of six-ring boron nitride in the (001) plane. The vectors of the direct lattice ($\tau_1$ and $\tau_2$) are plotted in dark green and the vectors of the reciprocal lattice ($R_1$ and $R_2$) are plotted in red. The edges of the Brillouin zone are shown as light green lines. The black lines represent the orientations of the vectors $x$ and $y$ of the international basis set representation for the direct lattice and for their analogs $k_x$ and $k_y$. Note the positions of the specific points of the reciprocal lattice $\Gamma$, $K_i$, and $M_i$.

Recent ellipsometry measurements recorded worldwide at room temperature confirm a broad reflectivity feature near the value of the fundamental bandgap [71] with some puzzling discrepancies attributed to differing amounts of hBN and turbostratic BN, and/or the co-existence of $sp^2$ and $sp^3$-bonded crystallites [118].

In Figure 8, we reproduced the low-temperature reflectivity of a bulk crystal with pure AA' stacking. This is restricted to the energy range of the direct bandgap [119], and it is complementary to the recent room temperature investigation in the broad spectral range [120]. In addition to the feature at 6.125 eV which probes the value of the fundamental direct bandgap at the *K* point of the BZ, phonon-assisted transitions are also evident that feature the low-energy wing of the main reflectance structure. Details about the in-depth interpretation of this reflectance experiment can be found in [119]. We believe that a uniaxial stress effect can just slightly enhance the reflectance feature of super broad intrinsic origin at the energy of the direct bandgap of hBN. When the length of y increases, the lengths $\Gamma M_2$ and $\Gamma M_5$ increase and the lengths of $\Gamma M_1$, $\Gamma M_3$, $\Gamma M_4$, and $\Gamma M_6$ decrease. Indirect transitions involving the series $K_i$ and $M_i$ points of the BZ of the stressed crystal will be broadened and thus will smoothen beyond the possibility to detect them.

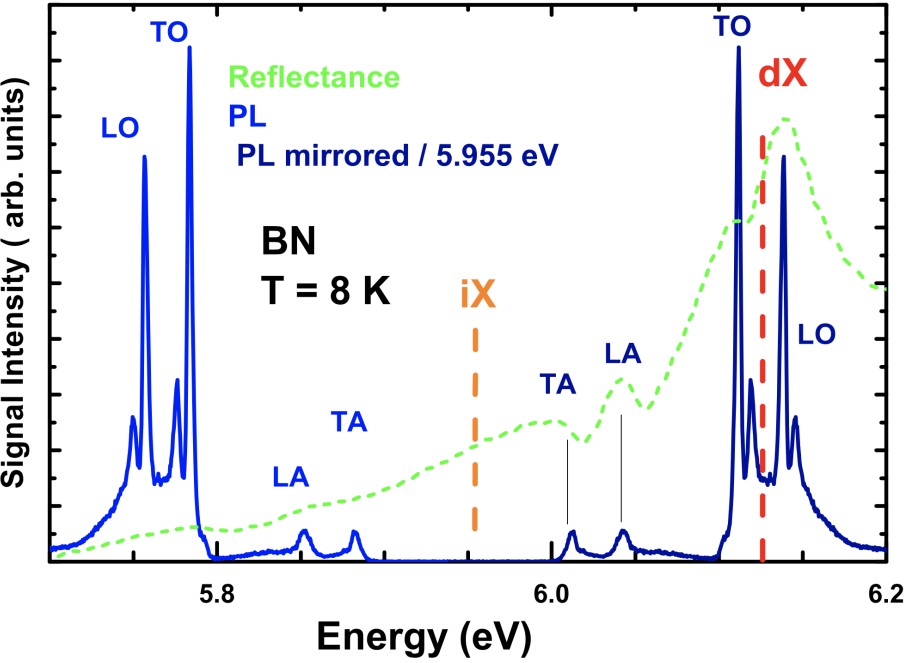

**Figure 8.** In blue is plotted the low-temperature photoluminescence spectrum of an ultra-pure hBN bulk crystal. In navy is plotted its mirror image relative to the energy (dashed orange line) of the indirect exciton (iX) at 5.955 eV. The reflectivity feature (green dashed line) displays a strong and broad feature at the energy (dashed red line at 6.125 eV) of the direct exciton dX. The series of lower energy features that are, versus energy of iX, one by one the mirror image of another one, is the evidence of indirect transitions that are observed thanks to the absorption of emissions of phonons [105,106,119].

## 3. The Optical Signatures of the Different Polytypes in the Deep Ultraviolet

### 3.1. Linear Optical Properties of hBN

Several authors have examined the problem of the determining of the bandgap of the different polytypes in the absence of excitonic effects or including them [26,43,57,62,104,121–128]. All studies predict different optical properties, in terms of photoluminescence energies, energies of singularities and their shapes, for the different stackings. The blue spectrum in Figure 8 reports a series of phonon-assisted transitions that are shifted from the energy of the forbidden indirect exciton relative to the energy of one phonon at the middle distance from the edge of the BZ. The band structure of hBN is indirect between the minimum of the conduction band at *M* and the top of the valence

band in the vicinity of $K$ [129]. It requires a phonon of the BZ at the middle of the $\Gamma - K$ direction to fulfill momentum conservation [130]. These phonon-assisted features are labeled in terms of LO, TO, LA, TA, $ZO_1$, and $ZO_2$ phonon replicas. The shape of the PL is ruled by selection rules [24,131] and the configuration of the experiment. At the low-energy side of the main peak of each phonon-assisted radiative recombination, there are overtones corresponding to a series of emissions associated with the low-energy Raman active mode $E_{2g}$ (at about 7 meV) [132]. Observing these overtones requires high-quality material; it is not possible for the AA stacking, as this low-energy Raman active mode does not exist as discussed earlier. At even lower energy, that is to say below 5.65 eV in hBN, there is emission involving one or more complementary TO($K$) phonons. This is an intervalley scattering process that can be a priori observed at specific energies for all polytypes. High-quality BN material is free from such contributions, as they require specific extended defects to furnish to the Fermi golden rule the final density of states that controls the intensity of light emission at these energies [6,100,104,133]. The presence or absence of such signatures in the photoluminescence (PL) or cathodoluminescence (CL) spectra can theoretically be used to reveal whether there is a given polytype or another one as a contaminant in the host matrix.

### 3.2. Linear Optical Properties of bBN

To stimulate the growth of different bulk crystals of BN polytypes, we have modified our growth protocol by adding a substantial amount of graphite to the molten Cr + Ni eutectic, which is used to grow BN by a reversed solubility method. Impurities can induce non-bulk (different from AA′) stacking [134,135]. The Bernal staggered stacking of the graphitic layers is expected to impact the precipitation of a different polytype of BN out of the molten ingot saturated in B and N when cooling it down. Luckily, we obtained (001)-oriented BN crystalline flakes with slightly different X-ray diffraction features compared to high-quality hBN. Namely, broader features associated to the (001) planes and complementary diffraction peaks evidenced the inclusion of polytypisms with bBN. Unfortunately, the BN polytype cannot be determined from the morphology of the crystal. Another sample was grown in which 5% vanadium was added to the Cr + Ni eutectic solvent. Both samples grown with Cr + Ni + C and Fe + V exhibited Bernal-related PL, as we shall see later, but in contrast to the X-ray spectrum of the former, the latter did not contain misoriented polymorphic inclusions [136]. However, its X-ray diffraction spectrum is not as good as those of a high-quality hBN sample. In Figure 9, we gathered typical PL spectra recorded on BN samples, grown with Cr + Ni + C (green), Fe + V (red), and a very high-quality hBN sample grown using Fe only as a metal solvent [137] (blue). The astonishing feature is a new photoluminescence line at 6.035 eV in samples grown from Cr + Ni + C and Fe + V that we attribute to the signature of the direct bandgap of the AB stacking, i.e., bBN, and that is never detected in high-purity BN samples. To do so, we take advantage of theoretical calculations [123,124], which included an excitonic effect, and predicted that the bandgap of bBN is larger than that for hBN. It is also higher than the value for rBN.

The series of defect-related lines at energies below 6.5 eV in hBN appear slightly blue shifted in bBN compared to hBN. However, there is a strong overlap of the contributions of bBN and hBN. In other words, this indicates that the sample contains multiple polytypes. In Figure 10, we plotted in blue the PL of our test hBN sample and in red the PL of the sample containing bBN. A 1200 grooves/mm grating was used to record the experimental data. The several contributions, separated by 7 meV, in the main 6.035 eV feature reveal a not strictly homogeneous system. It is very reasonable to attribute this to strain-induced splitting of the direct transition, and/or to electronic complementary low-frequency Raman scattering, but we do not have a definitive interpretation to offer. To help understand the red PL spectrum, the spectrum of hBN was blue-shifted by 74 meV and plotted in gray. Then, it becomes possible to discriminate from the red spectrum the contributions of hBN from those of bBN. Concerning the latter, as the phonon energies are similar for these

polytypes, the value of the indirect exciton in bBN is located at 6.029 eV in the low-energy wing of the PL signature of the direct bandgap, about 6 meV lower. Then, the contributions of phonon-assisted transitions, associated with the indirect bandgap noted iXbBN-PT, where the PTs are phonons at the center of the Brillouin zone, can be identified. In addition, the emission of a zone center $LO_\Gamma$ is detected at energy dXBN-$LO_\Gamma$ at an energy at about 5.838 eV. Note that it is worthwhile attributing the 6.035 eV luminescence to the recombination at the bottleneck of the lowest polariton branch of the direct exciton polariton systems; thus, the value of the excitonic gap is higher than this value by an amount dictated by the magnitude of the longitudinal–transverse (LT) splitting between the branches of dispersion of the exciton–polariton. This LT splitting is huge in hBN [119], about 400 meV, and from the band structure calculations referenced above [57,62,121–125,127–133,136,137], there is no reason for it to be drastically different in bBN.

A more accurate determination requires reflectivity measurements similar to the one shown in Figure 9 with lineshape fitting of the spectrum to obtain the relevant parameters of this direct exciton. Pelini et al. [138] reported in the 4 eV range the existence of a few PL lines at 4.12 eV and 4.14 eV, energies higher than the energy of the line detected in hBN at 4.097 eV [139], implying a bBN crystalline matrix. This indicates that the description of the microscopic nature of the colored center acting as an efficient light emitter at 4 eV cannot ignore the layer stacking. This is especially true, as single-photon emitters operating at 4 eV have been demonstrated [8] in hBN. Nothing is known regarding the performances of their bBN analogs.

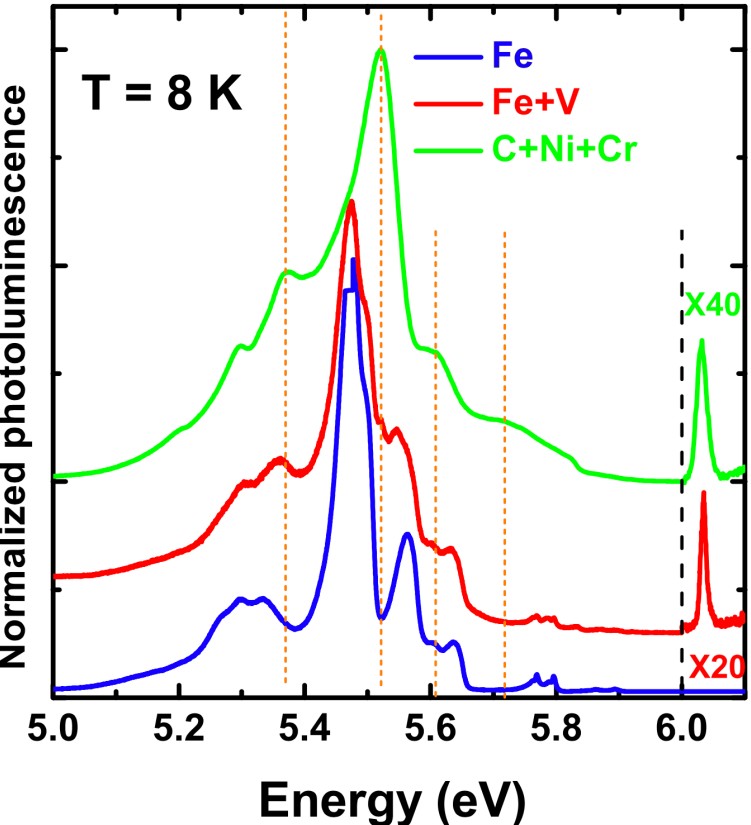

**Figure 9.** The 8 K macro-photoluminescence spectra in the 5–6.1 eV range of several samples showing the phonon-assisted transitions typical of hexagonal boron nitride (hBN; blue spectrum), mixed with the undulations linked to rBN and bBN inclusions impurities (green and red spectra). The signature of the direct bandgap of bBN is detected by a peak at 6.035 eV, following [136].

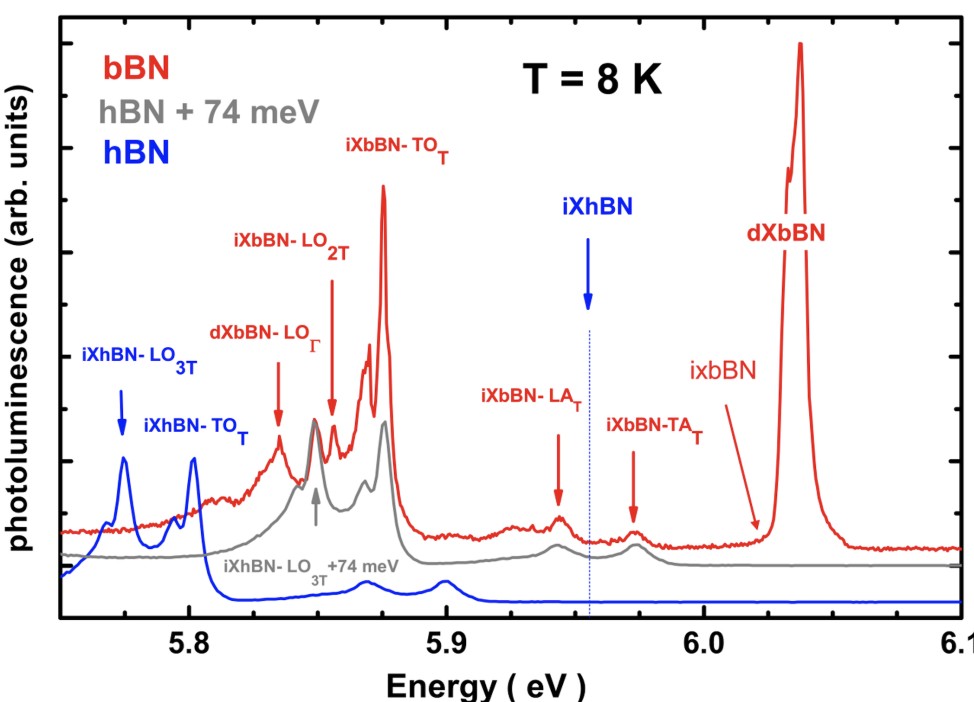

**Figure 10.** High resolution PL of a high-quality hBN crystal (blue) and of a piece of sp²-bonded BN with highly dominating bBN (red). The hBN PL is re-plotted in gray after being blue-shifted 74 meV. The energies of the indirect iXbBN and direct exciton dXbBN in bBN are indicated as well as the indirect exciton iXhBN in hBN. The different phonon replicas are also identified.

### 3.3. Linear Optical Properties of rBN

The optical properties of rBN have not yet been well documented except in Xu et al. [140]. X-ray diffraction experiments have revealed the presence of rhombohedral stacking in BN epilayers grown by chemical vapor deposition on several substrates, namely SiC, *c*-plane sapphire capped by GaN or AlN, and various orientations of the sapphire substrate [67–70,141]. To detect rBN stacking requires completing $\theta/2\theta$ Bragg–Brentano diffraction experiments by in-plane measurements, azimuthal scans ($\phi$-scans) and grazing incidence diffraction (GID). Rhombohedral BN films with twinned crystals that are rotated by 60° have been detected [67–70,72]. The photoluminescence of these deposited layers are dominated by two bands centered at 5.35 eV and 5.55 eV, respectively, and no analogs of the sharp phonon-assisted series of lines traditionally recorded can be measured. It is probably due to crystal twinning, which generates inhomogeneities and broadenings as well as transfers carriers to low energy traps. Interestingly, tBN has a PL signature that occurs as a complementary band at 5.4 eV, sandwiched between the preceding two, and with an intensity that follows the proportion of tBN relative to rBN in the crystals. The PL spectra measured in a couple of these rBN-rich epilayers are plotted in Figure 11 (red and wine) in addition to those typical of AA' (blue) and AB stackings (green) (after [72]). Similarly to what occurs for InSe crystals containing polytypism, there is a substantial overlap of the different optical signatures that may complicate the interpretation [142].

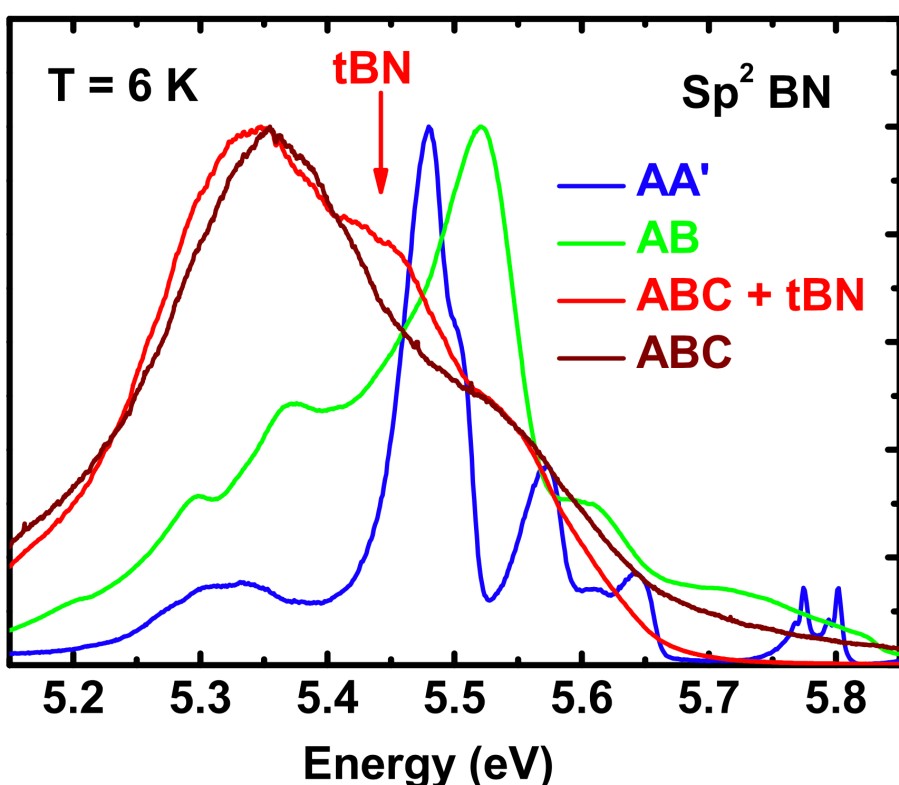

**Figure 11.** Photoluminescence spectra recorded on two rBN-rich epilayers (red and wine) in comparison with those typical AA' (blue) and AB (green) stackings. Note the signature of tBN. See reference [72] for more details.

### 3.4. Linear Optical Properties of the BN Monolayer

There has been a tremendous interest, soon after the demonstration of an indirect to direct bandgap crossover by Mak et al. [143], when decreasing the thickness of $MoS_2$ down to the monolayer, to reproduce this experiment in other 2D materials. There are several properties that changed from the bulk to the monolayer: the band structure itself and a strong renormalization of the exciton binding energy up to a huge value. This renormalization can be comparable in size to the value of the bandgap of the material in the absence of coupling of its electronic states with the electromagnetic field. This was established long ago through the theoretical predictions of Rytova [144] and Keldysh [145]. A more recent and detailed description of these mechanisms and their extension under the presence of disorder was published by E. V. Kirichenko and V. A. Stephanovich [146]. In BN, theoretical calculations including the Coulomb correlation predict a direct bandgap at *K* for the monolayer and an indirect or marginally direct bandgap for stackings of two or more layers [124,147–149]. Photoluminescence experiments [150–152] combined with reflectance measurements [150,151] have jointly confirmed using monolayers, deposited by molecular beam epitaxy on highly ordered pyrolytic graphite (HOPG) [150,151] and on suspended membranes [152], that it is a direct bandgap. The photoluminescence of the BN monolayer emits at 6.085 eV at low temperature, which was recently confirmed by the direct measurement of the density of states of a single monolayer of h-BN epitaxially grown on HOPG. This was achieved by performing low-temperature scanning tunneling microscopy (STM) and spectroscopy (STS) [153]. According to group theory, the $D_{3h}$ point group authorizes piezoelectric behavior, which was observed experimentally [154].

### 3.5. Second Harmonic Generation in Some Polytypes

Among layered BN's many interesting physical properties is its ability to demonstrate second harmonic generation in the polytypes without inversion symmetry [117]. The choice of ultraviolet emission by second harmonic generation (SHG) nicely integrates in the context of the quest for compact solid state deep ultraviolet emitters. The demonstra-

tion of SHG emission was achieved by shining light on a Cr + Ni + C sample, which had the Bernal stacking of layers (bBN or AB of spatial symmetry $P\bar{6}2m$ or $D_{3h}^3$), with a laser operating at 400 nm and observing light emission at 200 nm. To do so, we have examined a small piece of the sample described above by micro-photoluminescence spectroscopy at $T = 8$ K, which has a spatial resolution of 200 nm [133]. A photograph of a piece of BN unambiguously containing polytypism as can be evidenced by just using an optical microscope is shown in Figure 12a. Figure 12b is a map of the PL emission by this piece of sample, when illuminated with a laser wavelength of 196 nm. This is a composite containing a region of bBN (it emits light at 6.035 eV whilst hBN does not) sandwiched between two regions of hBN (for which the light emission does not occur at this energy, as is typical of hBN). We then shifted the laser light to 400 nm and recorded the light emitted with a wavelength of 200 nm at the strict position of the Bernal polytype (see Figure 12c). This experiment demonstrates second harmonic generation at 200 nm and makes Bernal BN a good candidate for devices susceptible of deep ultraviolet emission by up-conversion of a UVA light [155].

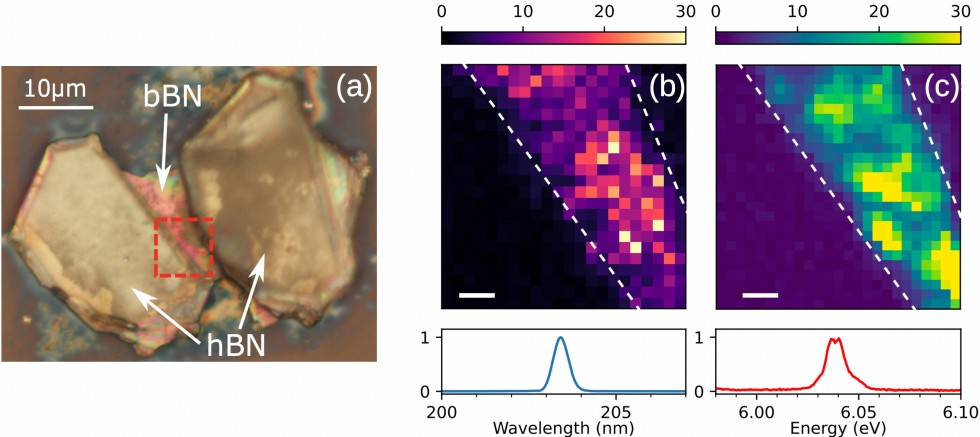

**Figure 12.** (**a**) Photograph of an exfoliated sample composed of a bBN region sandwiched between two hBN regions. (**b**) Mapping of the PL emission from the square area indicated in (**a**). (**c**) Mapping of the SHG emission. After [155].

## 4. Conclusions

Different layered boron nitride polytype crystals can be synthesized by tuning the growth conditions. In this review article, we emphasized that second harmonic generation of UVA light can produce an UVC emitter. There are not so many studies about the control of growth of layered BN with controlled polytypism, but it is obvious, having in mind the different applications of the different polytypes of silicon carbide, that such crystals can demonstrate very specific properties compared to hBN. The efficiency of SHG in the challenging area of the deep ultraviolet using Bernal boron nitride paves the way for its use in short-wavelength optoelectronics. Having in mind the SiC story, extension of our interest further than optoelectronics at short wavelengths might soon lead to other polytypes knocking at the doors of the device arena.

**Author Contributions:** B.G. wrote the paper. W.D. made all the theoretical calculations. M.M. performed the X-ray diffraction experiment. P.V. and G.C. supervised the photoluminescence measurements recorded by A.R. and the reflectivity measurements recorded by C.E. J.H.E. supervised the growth of boron nitride by J.L. and E.J. at KSU. All authors contributed to the interpretation of the experimental data. All authors have read and agreed to the published version of the manuscript.

**Funding:** This paper was financially supported by the network GaNeX (ANR-11-LABX-424 0014), the BONASPES project (ANR-19-CE30 0007), the ZEOLIGHT project (ANR-19-CE08-0016), and the Université de Montpellier. Support for BN crystal growth came from the Office of Naval Research, Award No. N00014-20-1-2474, and the National Science Foundation, Award No. CMMI 429 #1538127.

**Institutional Review Board Statement:** Not applicable.

**Informed Consent Statement:** Not applicable.

**Data Availability Statement:** Not applicable.

**Acknowledgments:** We are grateful to Sachin Sharma, Laurent Souqui, Henrik Pedersen and Hans Högberg for their epilayers with rBN polytypes. The photoluminescence spectra, which are published together with them in Moret et al. [72], are reproduced in Figure 11 and have been used to conceive the graphical abstract. We gratefully acknowledge T. Cohen and C. L'Henoret for their technical supports at the mechanics workshop.

**Conflicts of Interest:** The authors declare no conflict of interest.

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
