# Peer review of "Polytypes of sp2-Bonded Boron Nitride"

_crystals, doi:10.3390/cryst12060782_

Round 1
Reviewer 1 Report
The authors summarized some physical properties and the performance of deep ultraviolet emission under the special condition of second harmonic light generation for BN polytypes. The review described the polymorphism and polytypism of BN and further induced their optical signatures in the deep ultraviolet. The review can be considered for publication after addressing the following issues:
1. While summarizing and presenting the reported works, the authors should give their judgment and prospect for the development and application of BN in the field of deep UV.
2. The authors mentioned much research on the properties and applications of BN in the field of deep UV, and the difference and improvement between this work and similar review reports should be highlighted.
3. A summary figure reflecting the overall structure and ideas of the review should be added.
4. It is suggested to supplement some recent frontier works and integrate the scattered diagrams in this review.
Author Response
- While summarizing and presenting the reported works, the authors should give their judgment and prospect for the development and application of BN in the field of deep UV.
The answer to this point is obvious, otherwise we should not have written this review article.
- The authors mentioned much research on the properties and applications of BN in the field of deep UV, and the difference and improvement between this work and similar review reports should be highlighted.
There are no similar reports on that topics otherwise we should not have written it.
- A summary figure reflecting the overall structure and ideas of the review should be added.
We propose to illustrate there with the graphic abstract : that really snapshots our review article:
- It is suggested to supplement some recent frontier worksand integrate the scattered diagrams in this review.
We added this new reference that nicely bridges this review to applications complementary of those we had thought about
Après cette phrase: For deep UV range of emission/absorption, i.e. light with wavelengths near 200 nm, aluminum nitride and boron nitride (BN) are, among the III-V compounds, both candidates of choice [
Su-Beom Song, Sangho Yoon, So Young Kim, Sera Yang, Seung-Young Seo, Soonyoung Cha, Hyeon-Woo Jeong, Kenji Watanabe, Takashi Taniguchi, Gil-Ho Lee, Jun Sung Kim, Moon-Ho Jo and Jonghwan Kim, Deep-ultraviolet electroluminescence and photocurrent generation in graphene/hBN/graphene heterostructures Nature Communications volume 12, Article number: 7134 (2021) ].
Reviewer 2 Report
The paper entitled "Polytypes of sp2-bonded boron nitride" is a review paper. It is very well and clearly written. All the details of boron nitride, its challenges and progress are described. No doubt our semiconductor community is waiting for such a work. This work should be published in its present form, as is.
Author Response
Referee 2 had no remarks and did not ask for modifications.